# Quantifying Uncertainty in Answers from any Language Model and Enhancing their Trustworthiness

## Abstract

We introduce BSDETECTOR, a method for detecting bad and speculative answers from a pretrained Large Language Model by estimating a numeric confidence score for any output it generated. Our uncertainty quantification technique works for any LLM accessible only via a black-box API, whose training data remains unknown. By expending a bit of extra computation, users of any LLM API can now get the same response as they would ordinarily, as well as a confidence estimate that cautions when not to trust this response. Experiments on both closed and open-form Question-Answer benchmarks reveal that BSDETECTOR more accurately identifies incorrect LLM responses than alternative uncertainty estimation procedures (for both GPT-3 and ChatGPT). By sampling multiple responses from the LLM and considering the one with the highest confidence score, we can additionally obtain more accurate responses from the same LLM, without any extra training steps. In applications involving automated evaluation with LLMs, accounting for our confidence scores leads to more reliable evaluation in both human-in-the-loop and fully-automated settings (across both GPT 3.5 and 4).

## 1 Introduction

While the promise of Large Language Models (LLMs) and Agents (powered by LLMs) has become evident, their usage in high-value applications remains limited by their *unreliability*. Accessed via black-box APIs (via providers like OpenAI/Anthropic), today's best LLMs have been trained to produce convincing-looking responses and thus often appear overconfident (Ji et al., 2023). For many input prompts encountered in the wild, the model cannot be certain about the desired response (perhaps because the prompt is vague or is related to a specific fact/event absent from the training dataset), yet these models output plausible-sounding yet wildly incorrect answers in such scenarios. This *hallucination* problem has also plagued traditional supervised learning systems, where it is traditionally addressed via *uncertainty estimation* to know when one can trust a model's prediction (Gal & Ghahramani, 2016a; Lakshminarayanan et al., 2017; Guo et al., 2017; Liang et al., 2017; Fortunato et al., 2017; Gal & Ghahramani, 2016b; Kuleshov et al., 2018).

In traditional supervised learning, one has access to the training data of the model and its probabilistic estimates, as well as being able to modify the training procedure to improve model calibration (Gal & Ghahramani, 2016a; Fortunato et al., 2017). Other traditional uncertainty estimation procedures require the existence of a validation set that can be used for calibration (Angelopoulos & Bates, 2021). None of this is available for today's best LLMs, which may be given any imaginable prompt rather than (input, output) pairs stemming from a limited distribution. Thus approaches to uncertainty estimation for black-box LLMs must wrap the inference procedure.

Our proposed LLM uncertainty quantification technique, BSDETECTOR, calls the LLM API multiple times with varying prompts and sampling *temperature* values (see Figure 1). We expend extra computation in order to quantify how trustworthy the original LLM response is, a worthwhile tradeoff for high-stakes applications. Our method is conceptually straightforward, generally applicable across LLM providers (as well as Agent frameworks (Chase, 2022) or any stochastic text $\rightarrow$ text mapping), and produces confidence scores whose values are reliably lower for responses from the LLM that are more likely bad.

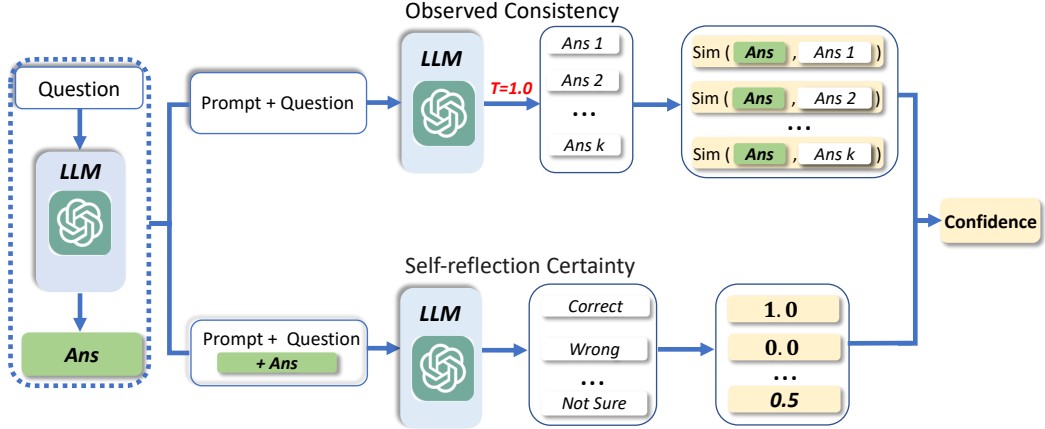

(a) Pipeline of BSDETECTOR, which can be applied to any LLM API. ($T = 1.0$ means temperature sampling with parameter 1.0, Sim $(\cdot,\cdot)$ means the semantic similarities between two sentences.)

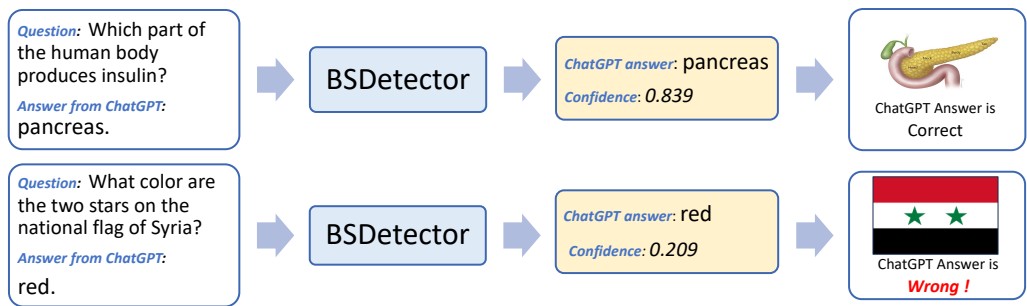

(b) Two prompts from a Trivia Q&A dataset (Joshi et al., 2017) and the responses from ChatGPT, along with the associated confidence scores from BSDETECTOR.

Figure 1: Overview of our LLM uncertainty quantification technique.

BSDETECTOR confidence scores allow LLMs to be more safely used in high-stakes applications, since we can know which LLM outputs are not to be trusted. Depending on the application, we can adaptively ask a human for an alternative response when the confidence score is low, automatically route the prompt to an alternative LLM provider, or simply respond "*I don't know*" when a confident response cannot be generated. Our experiments reveal that for Question-Answering applications, we can automatically generate *more accurate* answers by sampling multiple responses from the same LLM and selecting the response whose BSDETECTOR confidence estimate is the highest.

This paper primarily focuses on *Question-Answering* applications, but our same uncertainty estimates can also be applied to estimate how confident the LLM is in its response to a more general prompt. Intuitively, we'd like to see a low confidence score when the LLM outputs: a factually incorrect response to a question, a inaccurate summary requested for a document, or a generated article/message that semantically differs from the intention of the original request. Ensuring this is challenging without control over LLM training, but we can hope that in each of these three scenarios where the model generated a bad response, a well-trained LLM was also likely to output alternative responses (which more closely reflect the desired response). BSDETECTOR is baseed on this intuition, and is observed to produce effective uncertainty estimates with today's top LLMs from OpenAI across prompts from closed and open domain benchmark datasets.

## 2 RELATED WORK

**Uncertainty Estimation in Supervised Learning.** Understanding and quantifying the uncertainties associated with model predictions in traditional supervised learning has a rich history (Angelopoulos & Bates, 2021). Even when trained in a supervised manner, neural network models pose a unique set of challenges for measuring uncertainty and improving calibration (Papadopoulos et al., 2001; Riquelme et al., 2018). Much of this work stems from the field of computer vision, where distinct frameworks have been proposed by: Blundell et al. (2015); Gal & Ghahramani (2016a) to approximate Bayesian inference, Lakshminarayanan et al. (2017); Jain et al. (2020) to rely on straightforward deep ensembles, Liang et al. (2017); Papernot & McDaniel (2018) to detect Out-of-Distribution training samples. Parallel ideas for uncertainty estimation with supervised neural works have been developed in natural language processing (Fortunato et al., 2017; Gal & Ghahramani, 2016b; Kuleshov et al., 2018). However, these techniques are not directly applicable to today's best LLMs which are behind black-box APIs with unknown training data.

**Uncertainty Estimation for LLMs.** For estimating the confidence levels tied to responses output by large language models, Kuhn et al. (2023) introduce *semantic entropy*, incorporating linguistic invariances created by shared meanings. However their approach requires access to token-level probabilities from the LLM, which is often not accessible with today's black-box APIs. Kadavath et al. (2022) prompt the models to self-evaluate their answers and directly ask the LLM to produce the likelihood $P(\text{Answer is True})$ – also fine-tuning the model to output better values for its stated likelihood. Relatedly, Lin et al. (2022) prompt LLMs to generate both an answer and a level of confidence. Manakul et al. (2023) propose a sampling-based approach to detect hallucinated facts. All of these aforementioned approaches train additional models via supervised learning, unlike BSDETECTOR which does not employ any additional training. More recently, Tian et al. (2023) conduct evaluations of computationally feasible methods to extract confidence scores from the probabilities output by LLMs trained via Reinforcement Learning with Human Feedback. Lin et al. (2023) differentiate between *uncertainty* and *confidence* estimation for LLMs (under their terms, our work is focused on the latter, but without requiring access to the auto-regressive token probability estimates their method is based on). The works of Tian et al. (2023) and Lin et al. (2023) only study limited tasks, and it remains unclear whether their conclusions still hold in the context of reasoning or arithmetic. Here we demonstrate that our method produces effective uncertainty estimates across multiple domains involving reasoning, arithmetic, and knowledge of facts.

## 3 BSDETECTOR UNCERTAINTY ESTIMATION

When posing a *question* to LLMs, we aim to to estimate how confident we should be that a particular LLM *answer* is correct (or simply "good" for more general LLM responses). Specifically, for input question $x$, we want to not only obtain an answer $y$ from the LLM, but also an associated confidence score for this answer $C(x, y)$. Our confidence assessment derives from two factors: **Observed Consistency** and **Self-reflection Certainty**, which respectively are extrinsic and intrinsic evaluations of LLM confidence. Since a well-trained LLM should consider multiple different answers when asked an under-specified question or about something not contained in its training data, Observed Consistency extrinsically measures whether the LLM finds multiple contradictory answers likely to be good responses. Since effective LLMs can reasonably evaluate text from arbitrary agents, Self-Reflection Certainty directly asks the LLM to intrinsically reflect on whether its own previously-generated answer seems correct and how confident it is about this.

### 3.1 OBSERVED CONSISTENCY

The first critical measure of model uncertainty is contradiction score amongst possible answers LLMs gives to a particular input questions. Observed Consistency is an extrinsic confidence assessment performed by a user who engages in repeated interactions with LLMs. If a model exhibits strong observed consistency, it's less likely to present alternative responses that are substantially different from its initial answer. The idea was initially inspired by *Self-Consistency* (Wang et al., 2022). While Self-Consistency enhances LLM accuracy in closed-form tasks like arithmetic or commonsense reasoning, it falls short when applied to open-form tasks. Within the Self-consistency approach, an indicator function is used to measure the similarity amongst various likely responses. Here we

extend the indicator function to a particular form of semantic similarity based on contradiction ratings, enabling our approach to be used in both open and closed form tasks.

**Producing Diverse Output.**   Our first action runs the LLM multiple times to produce multiple varied responses. Besides increasing the temperature values (which can only be done so much without getting nonsensical outputs), we can alternatively modify the prompt itself when sampling each response to get a more diverse set of responses for computing the observed consistency. Here we add a Chain-of-Thoughts (CoT, Wei et al. (2022)) modification, along with other guidelines for output formatting, to the prompt used to sample these outputs. The specific prompt template is illustrated in Figure 6a, the outputs produced by this prompt are denoted as $\{\boldsymbol{y}_1, \boldsymbol{y}_2, ..., \boldsymbol{y}_k\}$, where $k$ is the number of sampled outputs. Higher values of $k$ lead to better uncertainty estimates, but require more computation (we found $k = 5$ works well enough in practice).

Note here we only modify the prompt used to sample varied responses for computing the observed consistency, *not* the prompt originally given to produce the original reference response. We tried alternative prompt modification techniques to encourage greater output diversity (such as adding additional made-up context in the prompt, or encouraging the LLM to answer as a specific persona), but found the CoT modification to work best (Table 3b).

**Measuring Similarity between Sampled and Original Answer.**   After receiving multiple outputs, the following step is to measure the similarities between each element in $\{\boldsymbol{y}_1, \boldsymbol{y}_2, ..., \boldsymbol{y}_k\}$ and original answer $\boldsymbol{y}$. Instead of using the indicator function to precisely match two numeric responses (e.g., 1.0 v.s. 2.0) or two choices (e.g. A v.s. B), we consider semantic similarities. Not just overall similarities (e.g. via LLM embeddings) which are sensitive to variation that does not necessarily indicate the LLM is uncertain, but rather measuring whether the semantics of the two outputs contradict one another or not. A common strategy to estimate this is to use a natural language inference classification system (NLI) (Kuhn et al., 2023), which classifies a pair of two text statements $\boldsymbol{y}_i$ and $\boldsymbol{y}$ as one of: *entailment*, *neutral*, or *contradiction*. Specifically, the input of NLI is formed by concatenating $\boldsymbol{y}_i$ and $\boldsymbol{y}$, and then NLI returns the probabilities $p$ for each of these 3 classes. For each element in $\{\boldsymbol{y}_1, \boldsymbol{y}_2, ..., \boldsymbol{y}_k\}$, we can get the similarity scores with respect to the original reference answer $\boldsymbol{y}$, denoted as $\{s_1, s_2, ..., s_k\}$.

Note that today's best NLI models (He et al., 2020) are significantly smaller than LLMs, and thus the NLI computation to obtain $s_i$ is negligible compared to sampling each LLM answer $\boldsymbol{y}_i$. However, even the best NLI models were trained on a limited dataset and thus do not always generalize reliably to arbitrary pairs of statements. In particular, we note the contradiction probabilities can be unreliable for single-word statements as encountered in certain closed-form tasks whose answers are likely not well-represented in the original NLI training dataset. To account for this, we additionally incorporate the indicator function in our similarity measure to enhance its stability for closed-form tasks. The indicator function is denoted as $r_i = \mathbb{1}[\boldsymbol{y} = \boldsymbol{y}_i]$ for $i = 1, 2, ..., k$.

For each element $\boldsymbol{y}_i$ in $\{\boldsymbol{y}_1, \boldsymbol{y}_2, ..., \boldsymbol{y}_k\}$, we derive the similarity score as:

$$o_i = \alpha s_i + (1 - \alpha) r_i \qquad (1)$$

Here $0 \leq \alpha \leq 1$ is a trade-off parameter (fixed at 0.8 in our experiments). It should have larger value the more we trust our NLI model to properly generalize its contradiction estimates. Finally, we average over $k$ samples to obtain the Observed Consistency score for answer $\boldsymbol{y}$ is $O = \bar{o}_i$.

### 3.2  SELF-REFLECTION CERTAINTY

Our *Self-reflection certainty* is an confidence estimate output by LLM itself when asked follow-up questions encouraging it to directly estimate the correctness of its original answer. Unlike sampling multiple outputs from the model (as in Observed Consistency) or computing likelihoods/entropies based on its token-probabilities which are *extrinsic* operations, self-reflection certainty is an *intrinsic* confidence assessment performed within the LLM. Because today's best LLMs are capable of accounting for rich evidence and evaluation of text (Kadavath et al., 2022; Lin et al., 2022), such intrinsic assessment via self-reflection can reveal additional shortcomings of LLM answers beyond extrinsic consistency assessment. For instance, the LLM might consistently produce the same nonsensical answer to a particular question it is not well equipped to handle, such that the observed consistency score fails to flag this answer as suspicious. Like CoT prompting, self-reflection allows

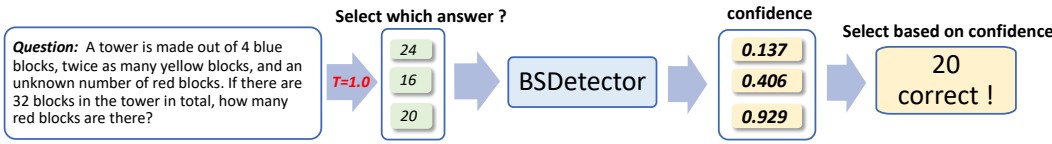

Figure 2: ChatGPT is used to generate the answers to arithmetic problem "A tower is ..." with temperature sampling $T = 1.0$. Subsequently, BSDETECTOR is utilized to select the most confident answer from the three possible answers.

the LLM to employ additional computation to reason more deeply about the correctness of its answer and consider additional evidence it finds relevant. Through these additional steps, the LLM can identify flaws in its original answer, even when it was a high-likelihood (and consistently produced) output for the original prompt.

To specifically calculate self-reflection certainty, we prompt the LLM to state *how confident* it is that its original answer was correct. Like Peng et al. (2023), we found asking LLMs to rate their confidence numerically on a continuous scale (0-100) tended to always yield overly high scores ($> 90$). Instead we ask the LLM to rate its confidence in its original answer via multiple follow-up questions each on a multiple-choice (e.g. 3-way) scale. For instance, we instruct the LLM to determine the correctness of the answer by choosing from the options: A) Correct, B) Incorrect, C) I am not sure. Our detailed self-reflection prompt template can be viewed in Figure 6b. We assign a numerical score for each choice: $A = 1.0$, $B = 0.0$ and $C = 0.5$, and finally, our self-reported certainty $S$ is the average of these scores over all rounds of such follow-up questions.

### 3.3 OVERALL CONFIDENCE ESTIMATE

Considering the distinct characteristics of the Observed Consistency and Self-reflection Certainty, we anticipate they might complement each other. BSDETECTOR aggregates the Observed Consistency and Self-reflection Certainty values into an overall confidence score for the LLM response:

$$C = \beta O + (1 - \beta)S \tag{2}$$

Here $0 \leq w_2 \leq 1$ is a trade-off parameter (fixed as 0.7 in our experiments). It should have larger value the more we trust the LLM's ability to do calibrated self-reflection assessment of arbitrary (question, answer) pairs.

## 4 APPLICATION: GENERATING MORE RELIABLE ANSWERS FROM ANY LLM

One straightforward application of our BSDETECTOR uncertainty estimate is to apply it to (each of) multiple candidate answers produced from the same LLM: $\{\boldsymbol{y}'_1, \boldsymbol{y}'_2, ..., \boldsymbol{y}'_k\}$ (including the original reference answer $\boldsymbol{y}$ in this set). This assessment allows is to determine which candidate LLM answer $\boldsymbol{y}'_i$ appears most trustworthy, and return that one instead of always returning $\boldsymbol{y}$ (see Figure 2). Specifically, we use the same prompt to ask the LLM to produce several responses via temperature sampling. For each candidate answer, we reuse the same set of previously-described LLM outputs $\{\boldsymbol{y}_1, \boldsymbol{y}_2, ..., \boldsymbol{y}_k\}$ to compute an observed-consistency score (reducing the computation required to assess the trustworthiness of a set of candidate answers). Following the standard BSDETECTOR procedure, we prompt the LLM to assign a self-reflection certainty to each candidate response. Finally we select the answer with highest BSDETECTOR confidence score, which is often the original reference answer $\boldsymbol{y}$, but not always. An alternate answer $\boldsymbol{y}'_i \neq \boldsymbol{y}$ can be deemed most trustworthy via this procedure only if: the LLM was able to identify fewer likely answers that contradict $\boldsymbol{y}'_i$ and was more certain about the correctness of $\boldsymbol{y}'_i$ during the intrinsic self-reflection assessment.

## 5 APPLICATION: MORE RELIABLE LLM-BASED (AUTOMATED) EVALUATION

In open-domain tasks, it is challenging to evaluate the correctness/quality of answers (irrespective of whether these answers were generated by a LLM or human). Often one resorts to automated

evaluation using models like GPT-3.5-turbo or GPT-4 to assess the correctness of answers (Lin et al., 2023; Taori et al., 2023; Xu et al., 2023). Recent instruction fine-tuning techniques such as Alpaca (Taori et al., 2023) and WizardLM (Xu et al., 2023) also utilize GPT-4 for automated evaluation of generated answers. Even when they are based on advanced LLMs like GPT-4, there remain **questions about the reliability of these LLM-based evaluations**.

Here we outline two ways to boost the reliability of LLM-based evaluation: *human-in-the–loop* and *fully automated*. Both start by computing BSDetector confidence scores for each LLM-evaluation (these scores estimate not the trustworthiness of the generator of the answers, but rather the evaluator of their correctness). Let $\mathcal{A}$ denote the subset of answers where the corresponding LLM-evaluation had the lowest BSDetector confidence scores (indicating the automated evaluation for this answer is untrustworthy). The gold-standard for evaluating open-domain answers is human inspection, but this is costly. Under a limited labor budget, we can boost the reliability of LLM-based evaluation by having humans only inspect and provide evaluations for the answers in $\mathcal{A}$. In settings where this *human-in-the-loop* approach is not possible, an alternative *fully-automated* way to boost the reliability of LLM-evaluation is to simply omit the answers in $\mathcal{A}$ entirely from the evaluation-set.

## 6 EXPERIMENTS

### 6.1 CALIBRATION OF UNCERTAINTY ESTIMATES

**Datasets.** Our experiments consider numerous question-answering benchmarks listed below. For each example in each benchmark dataset, the true answer is known enabling us to precisely assess the accuracy of LLM responses. We study performance in: GSM8K (Cobbe et al., 2021) and SVAMP (Patel et al., 2021), datasets composed of grade school math word problems, Commonsense Question Answering (CSQA) (Talmor et al., 2019), a dataset requiring some level of reasoning, and TriviaQA (Joshi et al., 2017), an open-form trivia question dataset that gauges models' factual knowledge. Because TriviaQA is open-domain, the correct answers provided do not entail all valid solutions, so we also manually validated the accuracy of LLM-generated responses.

**Experiment details.** We experiment on two LLMs from OpenAI: Text-Davinci-003 and GPT-3.5 Turbo. The reference answer $y$ is always produced with the temperature set at 0. To evaluate the confidence of $y$, we use prompt in Figure 6a to generate $k = 5$ outputs (unless otherwise stated) with the temperature set at 1.0 (the highest value allowed by the OpenAI API), combined with the indicator function to compute the observed-consistency score. For self-reflection certainty, two follow-up questions in Figure 6b are used to assess the correctness of the answer $y$. As previously described, we combine the observed-consistency and self-reflection certainty to derive the final confidence score.

**Evaluation metrics.** Following Kuhn et al. (2023), we use Area Under the Receiver Operator Characteristic Curve (AUROC) to evaluate the quality of our uncertainty estimates. AUROC represents the likelihood that a correct answer selected at random will have a higher uncertainty score compared to an randomly chosen incorrect answer. A higher AUROC value is preferable, with an ideal AUROC rating being 1, whereas a random uncertainty estimate would yield AUROC $= 0.5$. To evaluate *generation quality* from the method to get better LLM answers in Section 4, we simply rely on the accuracy of LLM answers.

**Baseline Methods.** Our study also evaluates the following baseline uncertainty estimation methods: *Likelihood Based Uncertainty* calculates the joint log-probability of a sequence from the autoregressive estimator and normalizes it by the sequence length (Malinin & Gales, 2020). While it represents the typical way to estimate *aleatoric* uncertainty in traditional supervised learning and structured prediction Hendrycks & Gimpel (2017), this approach can only can be applied to Text-Davinci-003, since the GPT-3.5 Turbo API does not provide access to token-level probabilities from the model. *Self-reflection Certainty* and BSDETECTOR are introduced in Fig 1a. *Temperature sampling* is equivalent to BSDETECTOR without: CoT prompting, self-reflection certainty, and the indicator function term inside of the text-similarity metric.

**Results.** Table 1 presents the performance results for our various benchmark tasks and uncertainty estimation methods. Here BSDETECTOR significantly outperforms all baselines across datasets, revealing that confidence from BSDETECTOR well aligns with accuracy.

Table 1: AUROC achieved by different confidence scoring methods across various datasets.

| LLM | Dataset | Likelihood Based Uncertainty | Temperature Sampling | Self-reflection Certainty | BSDETECTOR |
|---|---|---|---|---|---|
| Text-Davinci-003 | GSM8K | 0.647 | 0.614 | 0.521 | **0.867** |
| | CSQA | 0.490 | 0.540 | 0.539 | **0.743** |
| | SVAMP | 0.668 | 0.653 | 0.619 | **0.936** |
| | TriviaQA | 0.708 | 0.769 | 0.653 | **0.828** |
| GPT-3.5 Turbo | GSM8K | - | 0.660 | 0.831 | **0.951** |
| | CSQA | - | 0.583 | 0.506 | **0.769** |
| | SVAMP | - | 0.671 | 0.839 | **0.927** |
| | TriviaQA | - | 0.689 | 0.655 | **0.817** |

## 6.2 GENERATING MORE RELIABLE ANSWERS FROM ANY LLM

In Table 2, we select the response with the highest confidence out of 5 generated responses as described in Section 4. For all tasks, BSDETECTOR can identify less accurate responses and notably improve LLM accuracy. Table 2 compares this approach against the original single answer $y$ generated by the LLM (with temperature set to 0), referred to as the *Reference Answer*. While answers produced via the BSDETECTOR filtering procedure from Section 4 require 10x as much LLM-inference computation as the Reference Answer, the consistent accuracy gain observed in Table 2 makes this worthwhile for high-stakes applications.

Table 2: Generating more reliable LLM answers. We show the accuracy of each set of answers for the dataset produced from the LLM with a particular method.

| LLM | Dataset | Reference Answer (%) | BSDETECTOR (%) |
|---|---|---|---|
| Text-Davinci-003 | GSM8K | 12.50 | **16.83** |
| | CSQA | 71.50 | **72.83** |
| | SVAMP | 65.67 | **70.00** |
| | TriviaQA | 69.80 | **70.50** |
| GPT-3.5 Turbo | GSM8K | 47.47 | **69.44** |
| | CSQA | 72.72 | **73.22** |
| | SVAMP | 75.30 | **82.00** |
| | TriviaQA | 73.50 | **76.00** |

## 6.3 MORE RELIABLE LLM-BASED (AUTOMATED) EVALUATION

We first investigate how reliable GPT-4 based evaluation is in practice. First we employ the Text-Davinci-003 model to produce answers for **TriviaQA** (Joshi et al., 2017). Subsequently, GPT-4 is given the question and generated answer (from Text-Davinci-003) and asked to designate the answer as correct or incorrect (see the Figure 6c for the specific evaluation prompt). Since ground-truth answers are available for TriviaQA, we can report the accuracy of GPT-4 based evaluation, which is only 83.67% in this setting (Figure 3a). Next, we try using GPT-4 to assess the quality of answers. For example, alpaca-eval (Yann, 2023) utilizes GPT-4 to discern which answer from two LLMs is superior but it is unknown how reliable GPT-4 judgements are in their application. To investigate this, we consider a similar task: **Summarize-from-feedback** (Stiennon et al., 2020). This dataset provides the original context, a summary derived from that context, and a human assessment of the summary's quality (which we hold out only for reporting purposes here). We employ GPT-4 based evaluation to automatically rate each summary's quality, asking the LLM-evaluator to select from options: Bad, Fair, Good, or Excellent (see the Figure 6d for the specific evaluation prompt). Translating these

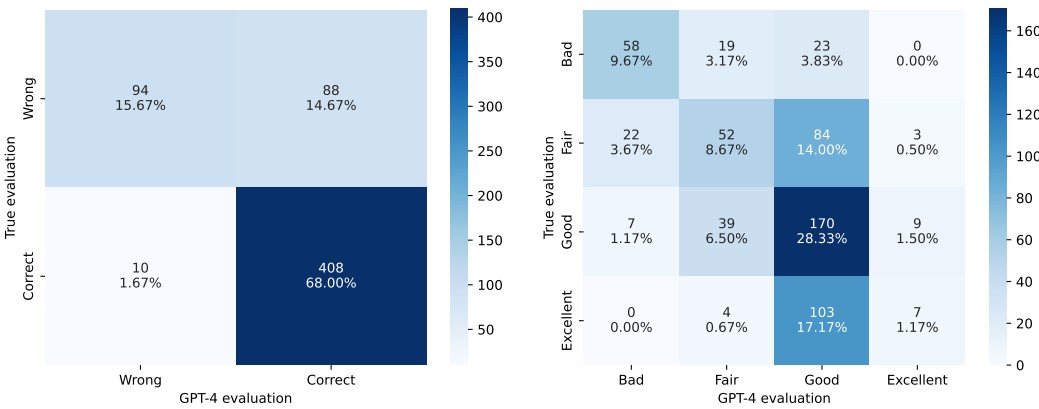

(a) TriviaQA: Overall GPT-4 accuracy: 83.67%     (b) Summarize-from-feedback: GPT-4 MSE: 0.707

Figure 3: Confusion matrix comparing automated GPT-4 evaluations vs. human evaluations.

ratings to a 1-4 numerical scale, we report the mean square error (MSE) between these automated GPT-4 ratings vs. the ground truth human ratings. Figure 3b shows this MSE is approximately 0.707. In both experiments, automated evaluation based on GPT-4 is not as reliable as one would hope to reach trustworthy conclusions.

Finally we study whether BSDETECTOR can help us achieve more reliable evaluations with GPT-4, as described in Section 5. We consider the TriviaQA and Summarize-from-feedback datasets with the same GPT-4 model and evaluation prompts from the previous paragraph, and compute BSDETECTOR confidence scores for the GPT-4 evaluator as described in Section 5. We first consider the *human-in-the-loop* setting, where a human provides the evaluation for answers in $\mathcal{A}$, defined as the subset of answers where the corresponding GPT-4 evaluation has BSDETECTOR confidence score amongst the $K$ lowest values. We compare the resulting set of combined automated + human evaluations (**confidence selection**) against a baseline set of combined automated + human evaluations, where the subset of answers evaluated by a human is chosen via **random selection** (rather than based on our confidence score). Figure 4 depicts the performance of the resulting *human-in-the-loop* evaluation vs. the number of answers $K$ evaluated by a human (remaining answers are all auto-evaluated by GPT-4). Across both datasets, guiding the human-the-loop evaluation based on BSDETECTOR confidence yields more reliable evaluations.

To conclude, we study the *fully-automated* approach to LLM-based evaluation from Section 5, which offers a labor-free way to utilize the BSDETECTOR confidence scores. Recall in this approach we simply omit the subset of answers in $\mathcal{A}$ from the evaluation-set entirely. We can then compute the average evaluation-score from GPT-4 as an overall quality estimate for the collection of generated answers. Intuitively, we do not want to include answers in this average whose GPT-4 evaluation is highly uncertain (to reduce variance), but discarding answers shrinks the remaining evaluation-set thus increasing variance of the resulting average.

Evaluating the impact of these variance changes requires statistical repetition, so we repeat the following procedure 500 times: For both datasets (TriviaQA, Summarize-from-feedback), we select 500 answers and calculate the average GPT4 evaluation-score over these answers. We call these the *full* dataset and the resulting average is the baseline score (estimator), whose accuracy/MSE we report against the average human evaluation score across the full dataset (estimand). To utilize BSDETECTOR for a more reliable estimator of the average human-evaluation score, we simply remove the 20% of answers with the lowest confidence scores for the corresponding GPT-4 evaluation, and compute the average GPT-4 evaluation score over the remaining 400 answers. As a sanity check, we also repeat this procedure but this time randomly dropping 20% of the answers (rather than based on confidence score), which purely increases the variance of resulting average GPT-4 evaluation score with no benefits. Figure 5 shows the resulting deviation between average GPT-evaluation score and average human evaluation score over all of these statistical replicate experiments. Across both datasets, we get more reliable average LLM-evaluation scores by discarding the answers with the lowest confidence scores for the corresponding LLM-evaluation. Preventing the high-uncertainty

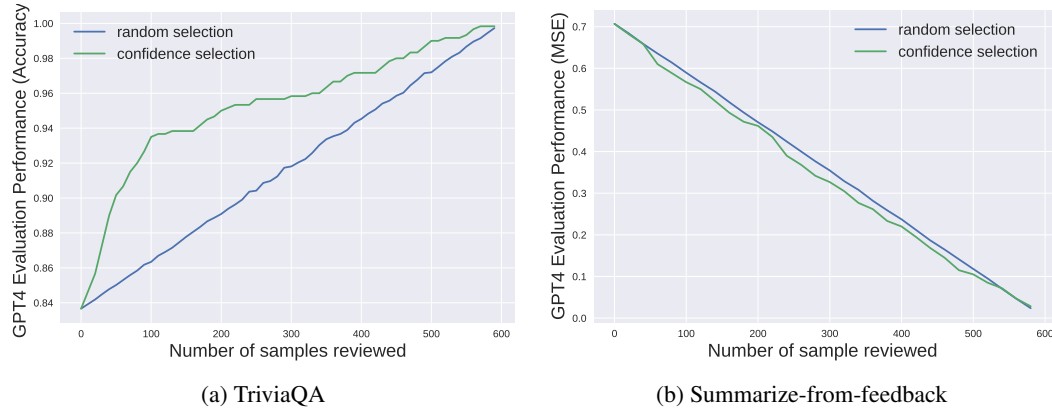

(a) TriviaQA

(b) Summarize-from-feedback

Figure 4: Human in the loop LLM-based evaluation, with the number of answers evaluated by humans varied along the x-axis (remaining answers are auto-evaluated by GPT-4). The resulting accuracy/MSE of the combined set of human + GPT-4 evaluations is shown along y-axis, under confidence-based vs. random selection to decide which subset of answers receive human evaluation.

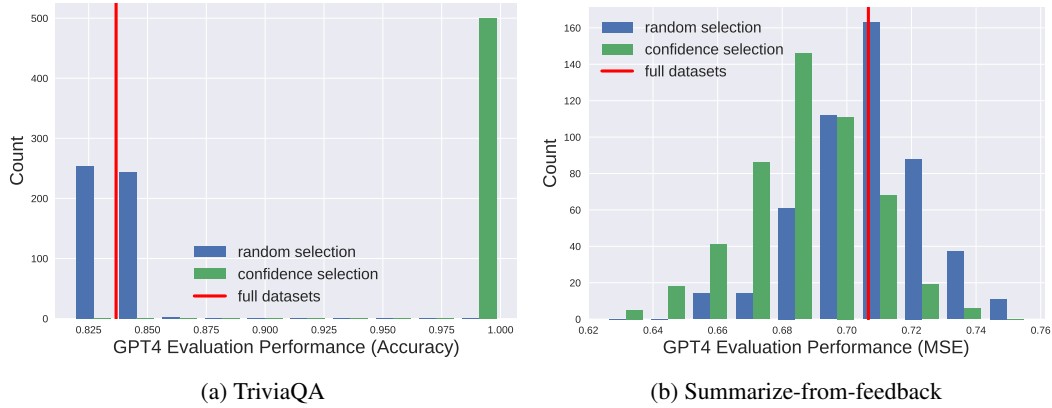

(a) TriviaQA

(b) Summarize-from-feedback

Figure 5: Fully-automated GPT-4 based evaluation, assessing the accuracy/MSE over many replicate datasets (observed counts amongst replicates on y-axis). By discarding the bottom 20% of evaluations with the lowest confidence, the average GPT-4 evaluation score consistently reaches an accuracy of 1.0 on TriviaQA, indicating completely trustworthy LLM-based evaluations (and the MSE of the average GPT-4 score consistently improves compared to the full dataset or discarding a random 20%).

LLM-evaluations from corrupting the average evaluation score is clearly worth the variance-penalty paid by shrinking the size of the evaluation set.

## 7 DISCUSSION

This paper presents BSDETECTOR, a method designed to identify unreliable or speculative answers from LLMs by computing a confidence score for its generated outputs. Our uncertainty estimates are applicable to any LLM, even those only accessible via a black-box API, and combine both intrinsic and extrinsic evaluations of confidence. By sampling multiple LLM answers and selecting the one with the highest associated confidence score, we can produce more accurate responses from the same LLM without any additional training. One open question is how to minimize the computational cost to achieve a desired level of confidence score calibration (for instance via adaptive produces that expend more compute only for assessing those answers whose confidence is hardest to estimate). Due to its simplicity and generality, we expect BSDETECTOR uncertainty estimation to find many applications across diverse domains/tasks, beyond the studies in this paper on: deciding what LLM responses cannot be trusted, and enhancing the accuracy of LLM answers and LLM-based evaluation.

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

# A APPENDIX

## A.1 DETAILS ABOUT NLI MODEL

Specifically, the input of NLI is formed by concatenating $\boldsymbol{y}_i$ and $\boldsymbol{y}$, and then NLI returns the probabilities $p$ for each of these 3 classes. Here we choose $1 - p_{contradiction}$ (output by an already trained NLI system (He et al., 2020)) as our similarity between two sampled LLM outputs. To mitigate positional bias within the NLI system, we consider both orders $(\boldsymbol{y}_i, \boldsymbol{y})$ and $(\boldsymbol{y}, \boldsymbol{y}_i)$, producing $1 - p_{contradiction}$ and $1 - p'_{contradiction}$ for each order and averaging these two values into a single similarity score. The similarity scores using NLI to assess each sampled LLM answer for contradictions with respect to the original reference answer are denoted:

$$s_i = \frac{1}{2}(1 - p_{contradiction} + 1 - p'_{contradiction}) \quad \text{for } i = 1, 2, ..., k.$$

## A.2 COMPUTE COSTS

The compute costs associated with various uncertainty methods differ. Uncertainty based on autoregressive likelihood is the most cost-effective, requiring only a single API call that returns the token-level probability. However, this cannot be implemented on GPT-3.5 Turbo since it does not provide token-level probabilities. While BSDETECTOR incurs a slight additional cost for self-certainty reflection in comparison to the baseline Temperature Sampling approach, Table 3a shows that even when we double the number of outputs from Temperature Sampling (thus allowing it far more compute than our approach), its performance remains inferior to BSDETECTOR.

## A.3 PROMPTS USED IN BSDETECTOR

Figure 6 show the prompts used in BSDETECTOR.

## A.4 ABLATION STUDY

In this section, we study that whether each component is required to achieve high quality. Our investigation leads to the following primary insights: 1) Enhancing the number of outputs and integrating CoT prompt in Observed Consistency result in a greater variety of responses, thereby making the confidence estimation more reliable. 2) Our similarity metric is crucial for capturing the variation between different responses.

### A.4.1 INCREASING THE NUMBER OF OUTPUTS AND INTEGRATING COT PROMPT INTRODUCE MORE DIVERSITY?

Table 3a shows an ablation study involving the number of outputs in Observed Consistency, we compare 5 and 10 outputs, observing that for each dataset 10 outputs outperforms 5 outputs. However, for GSM8K, SVAMP, and TriviaQA, the gain from 5 to 10 outputs is marginal. Given the trade-off between cost and performance, and considering that doubling the API calls results in only a slight improvement, we decide to stick with 5 outputs in our experiments. Table 3b indicates that CoT is essential for introducing the diversity of responses and achieving the good confidence estimation performance.

Table 3: Ablation study

(a) AUC of BSDETECTOR with different numbers of outputs.

|  | 5 outputs | 10 outputs |
|---|---|---|
| GSM8K | 0.951 | 0.961 |
| CSQA | 0.769 | 0.802 |
| SVAMP | 0.927 | 0.937 |
| TriviaQA | 0.817 | 0.814 |

(b) AUC of BSDETECTOR without and with CoT prompt augmentation.

|  | Remove CoT prompting | BSDETECTOR |
|---|---|---|
| GSM8K | 0.837 | 0.951 |
| CSQA | 0.665 | 0.769 |
| SVAMP | 0.882 | 0.927 |
| TriviaQA | 0.792 | 0.817 |

```
Please strictly use the following template to provide answer:
explanation: [insert step-by-step analysis], answer: [provide
your answer] + Question: [User Provided]
```

(a) Prompt template for Observed Consistency

```
1. Question: [User Provided], Proposed Answer: [User/LLMs
Provided]. Is the proposed answer: (A) Correct (B) Incorrect
(C) I am not sure. The output should strictly use the
following template: explanation: [insert analysis], answer:
[choose one letter from among choices A through C]

2. Question: [User Provided], Proposed Answer: [User/LLMs
Provided]. Are you really sure the proposed answer is
correct? Choose again: (A) Correct (B) Incorrect (C) I am
not sure. The output should strictly use the following
template: explanation: [insert analysis], answer: [choose
one letter from among choices A through C]
```

(b) Prompt template for Self-reflection Certainty

```
"Statement: " + [User Provided Question] + "\n" + "Response: " +
+ [User Provided Answer] + "\n" + "What do you think of this
response to the statement is correct or incorrect, please pick
one of these choices:"
```

(c) Prompt template for triviaQA in the application of using BSDETECTOR as an evaluator.

```
"Article: " + [User Provided Context] + "\n\n\n" + "Summary: " +
[User Provided Summary] + " Your task: Rate how well this
Summary overall represents the original Article? Choose from the
options: [Bad, Fair, Good, Excellent]. Bad indicates the Summary
is inaccurate, misses important information, or is incoherent
and hard to understand. Fair indicates the Summary has some flaw
in terms of accuracy, coverage, and coherence, but is otherwise
decent along the other dimensions. Good indicates the Summary
accurately matches the factual information, conveys the main
idea of the Article, and is easy to understand but has some
minor flaws in any dimensions. Excellent indicates it is hard to
find ways to make the Summary better. Your rating (chosen from
Bad, Fair, Good, Excellent):"
```

(d) Prompt template for Summarize-from-feedback in the application of using BSDETECTOR as an evaluator.

Figure 6: Prompts used to produce the confidence score in BSDETECTOR.

### A.4.2 EFFECT OF DIFFERENT SENTENCE SIMILARITY METRICS

Table 4 shows the AUC performance with different similarity metrics. We compare **Jaccard similarity** calculated by dividing the number of observations in both output strings by the number of observations in either string, **LLM-embedding** utilizing text-embedding-ada-002[1] to get embedding for each output answers and calculating the cosine similarities between them, **NLI** using an off-the-shelf

---

[1]https://platform.openai.com/docs/api-reference/embeddings

DeBERTa-large model (He et al., 2020) for the purpose of categorizing into one of: entailment, contradiction, and neutral, NLI (1-contradiction) using $1 - p_{contradiction}$ as the final similarities metrics. Table 4 shows that the similarity metric used in BSDETECTOR is essential for discerning the differences among various responses.

Table 4: Effect of different sentence similarity metrics

| Dataset | Jaccard | LLM-embedding | NLI (1-contradiction) | BSDETECTOR |
|---------|---------|---------------|------------------------|------------|
| GSM8K | 0.896 | 0.866 | 0.892 | 0.951 |
| CSQA | 0.857 | 0.849 | 0.727 | 0.769 |
| SVAMP | 0.917 | 0.888 | 0.901 | 0.927 |
| TriviaQA | 0.650 | 0.642 | 0.794 | 0.817 |

