# OpenReview forum: "Quantifying Uncertainty in Answers from any Language Model and Enhancing their Trustworthiness"
_ICLR.cc/2024/Conference — ICLR 2024 Conference Withdrawn Submission_

### Official Review · Reviewer_rA28 · 2023-10-31

**Soundness:** 2 fair
**Presentation:** 2 fair
**Contribution:** 2 fair
**Rating:** 3
**Confidence:** 4

**Summary:**

The paper presents a method designed to detect unreliable answers from LLMs by determining a numeric confidence score for their outputs. This technique is applicable to LLMs accessed through a black-box API, even if their training data is undisclosed. Compared to other uncertainty estimation methods, BSDETECTOR more proficiently spots incorrect answers from LLMs, including GPT-3 and ChatGPT. Furthermore, by selecting the response with the highest confidence score among multiple LLM outputs, the accuracy of the responses can be enhanced without additional training. When these confidence scores are incorporated, evaluations involving LLMs become more dependable in both human-involved and fully-automated scenarios.

**Strengths:**

1. The paper highlights the importance of uncertainty quantification for LLMs and presents a novel approach to analyzing the uncertainty. The idea is straightforward and effective: Multiple responses are sampled from the LLM, and the one with the highest confidence score is considered the most accurate response.
2. The method works for any LLM accessible only via a black-box API, without knowledge of the training data.
3. The method focuses on Question-Answering applications but mentions that the uncertainty estimates can be applied to other prompts as well.

**Weaknesses:**

1. The authors may overclaim the contribution of this paper. Specifically, the authors mentioned several times that the proposed methods can be applied to any LLMs on various tasks. However, the experiments only contain OpenAI's APIs' performance on QA datasets.
2. A large number of uncertainty quantification methods are missing, which makes the related works section incomplete and lacks a fair model assessment.
3. Letting ChatGPT explain itself seems quite straightforward. Since ChatGPT is trained with RLHF, it rarely outputs uncertain answers in ChatGPT's opinion. I feel like training a surrogate model to make the answer judgment is more intuitive.

**Questions:**

1. Other than AUROC, it would be also good to show the accuracy of predictions, which can give a basic sense of the confidence score.
2. What is $\omega_2$ in Sec. 3.3? Should it be $\beta$?
3. How do you find the value of $\alpha$ and $\beta$? Have you conducted any sensitivity analysis on hyper-parameters?
4. Could you give a formal definition of confidence selection and random selection in Sec. 6.3 with more details?
5. Finally, a large number of baselines are missing. e.g., [1] and [2].

[1] Stephanie Lin, Jacob Hilton, and Owain Evans. Teaching models to express their uncertainty in
words. arXiv preprint arXiv:2205.14334, 2022a.
[2] Lorenz Kuhn, Yarin Gal, and Sebastian Farquhar. Semantic uncertainty: Linguistic invariances for
uncertainty estimation in natural language generation. arXiv preprint arXiv:2302.09664, 2023.

---

> ### Author Response · Authors · 2023-11-22
>
> Thanks for your constructive comments, please refer to the `Response to all reviewers` for the two frequently asked questions. We also provide the answer to your other question:
>
> **Question**: Letting ChatGPT explain itself seems quite straightforward. Since ChatGPT is trained with RLHF, it rarely outputs uncertain answers in ChatGPT's opinion. I feel like training a surrogate model to make the answer judgment is more intuitive.
>
>
> **Answer:** It's true that ChatGPT's training through RLHF does typically lead to responses that seem certain in the model's 'opinion.' However, it's important to remember that this certainty does not always equate to **factual accuracy**, and the model can occasionally produce incorrect or 'hallucinated' answers. The idea of training a surrogate model for answer judgment is indeed an intuitive approach. It could provide an additional layer of evaluation, potentially enhancing the reliability of the responses. However, the development of such a model would involve complex challenges, particularly in creating a well-curated and extensive training dataset that accurately represents a wide range of possible responses and their corresponding judgments.
>
>
> **Question:** Regarding to various LLM.
>
> **Answer:** We acknowledge that our testing was limited to OpenAI's products. This is primarily because OpenAI's product are currently the most extensively used. We attempted to obtain access to other closed-source APIs like Bard from Google, but found that Bard's API is not as widely available to public, only open to limited number of users. Regarding open-source model like LLama, they typically return the probability for each token, allowing for different methods to quantify uncertainty, as mentioned by reviewer [1]. Our approach is designed for more general settings without token-probabilities available (closed APIs, chains/agents and other text->text mappings that are not merely one LLM input->output).  Note that the experiments with GPT3 show our method significantly outperforms relying on the token-probabilities to quantify LLM uncertainty.
>
>
>
> [1]. Semantic uncertainty: Linguistic invariances for uncertainty estimation in natural language generation. arXiv preprint arXiv:2302.09664, 2023.
>
>
>
> **Question:** Other than AUROC, it would be also good to show the accuracy of predictions, which can give a basic sense of the confidence score.
>
> **Answer:** We have already shown the accuracy of predictions in the table 2
>
> **Question:** Hyperparameters chosen in our method
>
> **Answer:** Sorry for the typo, $\omega_2$ is $\beta$. The chosen of $\alpha$ and $\beta$ is from the grid search in [0, 1], the chosen of temperature for $T = 1.0$ is simply that is the largest allowed temperature value in the API.
>
>
> **Question:** Could you give a formal definition of confidence selection and random selection in Sec. 6.3 with more details?
>
> **Answer:** If we have 10 prediction, Confidence selection means selecting  prediction where the confidence score (return by our method) exceeds a predetermined threshold. Random Selection, on the other hand, is a process where choices are made randomly, without any specific criteria or pattern

---

### Official Review · Reviewer_8Bjv · 2023-10-31

**Soundness:** 2 fair
**Presentation:** 1 poor
**Contribution:** 2 fair
**Rating:** 5
**Confidence:** 4

**Summary:**

This paper introduces a confidence score named "BSDETECTOR" which is designed to detect failures, improve accuracy by identifying answers with highest score, and evaluate the quality of LLMs' generation.

The "BSDETECTOR" score is derived from a linear combination of a consistency score and a self-reflection score. For the consistency score, multiple responses are generated by varying the prompt for the same question (as demonstrated in Figure 6 (a)). The confidence score for a specific response is then determined by calculating the average similarity between this response and the others. On the other hand, the self-reflection score is obtained by posing questions to the model such as, "Is the proposed answer: (A) Correct (B) Incorrect (C) I am not sure?" followed by, "Are you really sure the proposed answer is correct? Choose again."

To evaluate the performance, the author focuses on two tasks:
1) Failure prediction: Evaluating the performance of failure prediction on QA dataset using AUROC and ACC.
2) Active Learning: For the task of Summarize-from-feedback, the authors use LLM to assess the quality of the generated summaries. Then they have a budget to ask human to give their evaluation. Compared to random selection, selecting the K samples with the lowest "BSDETECTOR" scores for human evaluation results in a closer alignment with the ground truth evaluation (i.e., human evaluations provided by the dataset).

**Strengths:**

Originality/Significance: The author proposes a straightforward metric to assess whether the LLM's generation is reliable, which is a linear combination of consistency score and LLM self-evaluation score. Furthermore, the author evaluates its performance by applying it to the task of active learning.

**Weaknesses:**

**Originality:** The paper extends the framework of semantic uncertainty[1]: sampling multiple answers and computing their semantic similarity. Unlike the previous work that utilised the model's own log-likelihood to compute the final uncertainty, this paper leverages the average semantic similarity generated by NLI to indicate uncertainty. The rationale is that high similarity to other candidate answers implies the answer is more likely to be correct. Moreover, the authors also introduce a "self-reflection score" to enhance performance. The linear combination of consistency and self-reflection score is simple, but the paper lacks sufficient ablation studies to show their complementary nature, making this contribution appear somewhat limited.

[1]Kuhn, Lorenz, Yarin Gal, and Sebastian Farquhar. "Semantic uncertainty: Linguistic invariances for uncertainty estimation in natural language generation." ICLR 2023.

**Quality:** The experiments lack validating ablation studies: 1) what is the relationship between consistency and self-reflection score? Are the complementary or repetitive (i.e., the sample groups that they succeed and fail are the same)? What is the reason for specific weights in equations (like 0.7 for consistency in Eq. (2) and 0.8 in Eq. (1)), or settings (like T=1 for temperature sampling). Justifications for these choices are missing.

**Clarity:**

1. Several descriptions are ambiguous. For instance, what does "counts" in figure 5 refer to? Does mean the number of samples for each trial's dataset or is it the number of trails? Which dataset was Table 4 based on?
2. Some statements might be inaccurate. For instance, the statement about "AUROC represents the likelihood..." seems off. The correct logic should perhaps be about a "certainty score" rather than an "uncertainty score".
3. Some descriptions are hard to understand: e.g., the sentence "Note here we only modify the prompt used to sample varied responses for computing the observed consistency, not the prompt originally given to produce the original reference response" in Sec 3.1 "Producing Diverse Output". How do you modify the prompt? For Sec 3.2's "Through these additional steps, xxx" what does this sentence mean?
4. Necessary citations are missing in places like the introduction "any stochastic text → text mapping" (btw, what does this mean?) and Sec 3.2 "Like CoT prompting, self-reflection allows the LLM to ...".

**Significance:** The paper misses a discussion on the reliability of LLM's generation, especially in determining whether an answer from LLM is "good" or correct. It seems like the authors are relying on LLM's capability to distinguish good from bad by using description, which is questionable. If LLM's performance is weak, the improvement this method offers may be limited. The introduction of consistency suggests an ensemble approach, but its reliance on semantic similarity, which is dependent on an NLI model, is also limited, especially for long-sequence texts. This might explain the authors' choice of evaluating on datasets like "trivia qa" and "summarize from feedback" rather than the summarization itself.

**Questions:**

Refer to the above sections for other questions.
1. For the section titled "Sec 6.1 dataset", how exactly was the "we also manually validated the accuracy of LLM-generated responses" conducted?
2. In the last paragraph of "Sec 3.2": If you ask LLM, "Are you really sure the proposed answer is correct? Choose again", will this prompt introduce some bias? For instance, GPT-4 often exhibits sycophantic behavior[2]. If you continually question it on an issue, even if its response is correct, it might change it to an incorrect one. How do you address this issue?

[2] Liu, Yang, Yuanshun Yao, Jean-Francois Ton, Xiaoying Zhang, Ruocheng Guo Hao Cheng, Yegor Klochkov, Muhammad Faaiz Taufiq, and Hang Li. "Trustworthy LLMs: a Survey and Guideline for Evaluating Large Language Models' Alignment." _arXiv preprint arXiv:2308.05374_ (2023).

---

> ### Author Response · Authors · 2023-11-22
>
> Thanks for your constructive comments, please refer to the `Response to all reviewers` for the two frequently asked questions. We also provide the answer to your other question:
>
>
> **Question:** Several descriptions are ambiguous. For instance, what does "counts" in figure 5 refer to? Does mean the number of samples for each trial's dataset or is it the number of trails? Which dataset was Table 4 based on?
>
> **Answer:**" counts" in figure 5 refers to the number of trails. Table 4 is base on the TriviaQA and Summarize-from-feedback two datasets.
>
> **Question:** Some statements might be inaccurate. For instance, the statement about "AUROC represents the likelihood..." seems off. The correct logic should perhaps be about a "certainty score" rather than an "uncertainty score".
>
> **Answer:** AUROC represents the likelihood that a classifier will rank a randomly chosen positive instance higher than a randomly chosen negative one, assuming 'positive' ranks higher than 'negative'. Usually, we name it as "confidence score" instead of "certainty score".
>
> **Question:** Some descriptions are hard to understand: e.g., the sentence "Note here we only modify the prompt used to sample varied responses for computing the observed consistency, not the prompt originally given to produce the original reference response" in Sec 3.1 "Producing Diverse Output". How do you modify the prompt?
>
> **Answer:** we only modify the prompt, e.g., adding CoT prompt to **generate the observed consistency**. But for the reference answer, we do not modify any prompt, same with the user provided.
>
>
> **Question:** For the section titled "Sec 6.1 dataset", how exactly was the "we also manually validated the accuracy of LLM-generated responses" conducted?
>
> **Answer:** "how exactly was the "we also manually validated the accuracy of LLM-generated responses"" means we do some human check of the LLM-generated answer.  Since the correct answers provided by the datasets do not entail all valid solutions.
>
>
> **Question:** In the last paragraph of "Sec 3.2": If you ask LLM, "Are you really sure the proposed answer is correct? Choose again", will this prompt introduce some bias? For instance, GPT-4 often exhibits sycophantic behavior[2]. If you continually question it on an issue, even if its response is correct, it might change it to an incorrect one. How do you address this issue?
>
> **Answer:** Asking a Large Language Model (LLM) like GPT-4 to reconsider its answer with a prompt like "Are you really sure the proposed answer is correct? Choose again" can indeed introduce certain biases or lead to unintended consequences in the model's responses. Ensuring neutral phrasing and seeking elaboration (for example we also prompt LLM to output the explanation)  can help maintain the consistency and reliability of the model's outputs.

---

### Official Review · Reviewer_5FRv · 2023-10-31

**Soundness:** 2 fair
**Presentation:** 2 fair
**Contribution:** 2 fair
**Rating:** 3
**Confidence:** 4

**Summary:**

The paper proposes a method for quantifying the uncertainty of the answers of language models, in an opaque-box fashion. This means that the methods leverages alternative prompting techniques and multiple model calls to establish uncertainty but it does not require model weights or training data. The method uses observed consistency (established through multiple model calls at different temperatures on the same question) and self-reflection consistency (established through follow up model calls that clarify accuracy of an initial given answer). The method is evaluated by showing the performance of the uncertainty scores in predicting answer accuracy and also on its usefulness for improving model performance in general.

**Strengths:**

S1 - Uncertainty quantification is an important problem and even more relevant in the era of generative models

S2 - The notions of observed consistency and self-reflection consistency are fundamental and backed up by previous research and empirical evidence.

**Weaknesses:**

W1 - Experiments and results: It seems like the work uses standard prompting throughout the experiments and not chain of thought reasoning. CoT is necessary to see in all relevant results and even for uncertainty quantification for a fair comparison. For example, in Table 2 authors show how their technique can improve model performance but then in many of those datasets CoT offers a better result based on previous work. Similar techniques to cot can be used to also have the model report its own uncertainty and reasoning within the same model call.

W2 - Presentation: The authors can consider toning down the claims in the paper, with respect to the applicability of the results and breadth. There are several points in which the results are overly claimed in a misleading way that can convey confusing information to the reader. For example,

E1- "This paper primarily focuses on Question-Answering applications, but our same uncertainty estimates can also be applied to estimate how confident the LLM is in its response to a more general prompt." - To make such a claim one needs to evaluate and adapt the method such that it applies to longer generations and estimate uncertainty on different parts of the text. This is non trivial and still an open problem for longer generations, but the way how this paragraph is phrased makes it seem like this is straightforward.

E2- "Section 6.2 While answers produced via the BSDETECTOR filtering procedure from Section 4 require 10x as much LLM-inference computation as the Reference Answer, the consistent accuracy gain observed in Table 2 makes this worthwhile for high-stakes applications." The accuracy gains in Table 2 are in many cases modest, and in other cases not competitive with Chain Of Thought results presented in several previous works. https://github.com/FranxYao/chain-of-thought-hub is an example of results but similar results have been reported in papers discussing standard cot, tree of thought, diversity of thought, and self consistency

W3- Relationship to related work and novelty. In overall, the paper could do a better job in connecting the work with previous results and clarifying novelty with respect to these. See below:

Self-contradictory Hallucinations of Large Language Models: Evaluation, Detection and Mitigation
https://arxiv.org/abs/2305.15852v1
This is relevant to self-reflection consistency

Self-consistency improves chain of thought reasoning in language models https://arxiv.org/abs/2203.11171
This is relevant to observed consistency

**Questions:**

Q1- For future revisions, authors could consider clarifying for each figure and table what type of evaluation was used (manual human annotation on evaluation, automated, llm-based, etc). It is sometimes hard to follow which parts were evaluated through other models (or not)

Q2- Section 3.3. Does w2 refer to beta here? How was this parameter fixed to 0.7? Does this mean that self-reflection consistency is more useful for the models in the study?

**Details Of Ethics Concerns:**

BSDETECTOR could stand for Bull**t detector (not sure if this was the intention). The authors could consider changing the name to something more scientific and more relevant to uncertainty terminology.

---

> ### Author Response · Authors · 2023-11-22
>
> Thanks for your constructive comments, please refer to the `Response to all reviewers` for the two frequently asked questions. We also provide the answer to your other question:
>
> **Question:** CoT is necessary to see in all relevant results and even for uncertainty quantification for a fair comparison.
>
> **Answer:** We include Chain of Thought (CoT) prompting for answer generation. For each question, we use CoT prompting to generate 10 candidate answers. Each of these answers is accompanied by a confidence score, allowing us to select the response with the highest confidence. Here are the results.
>
>
> | Dataset  	| CoT prompting 	| BSdetector with CoT prompting (our method)|
> |----------|---------------|--------------------------------------------|
> | GSM8K    	| 71            	| 73                                         	|
> | CSQA     	| 74            	| 76                                         	|
> | SVAMP    	| 78            	| 83                                         	|
> | TriviaQA 	| 75            	| 79                                         	|
>
> Our method is better than CoT prompting. It's important to clarify that the dataset utilized in our study differs from the one in https://github.com/FranxYao/chain-of-thought-hub. For instance, while their GSM8k dataset contains 8,000 samples, we employed a version of GSM8k with only 1,318 samples as referenced in the CoT paper (https://arxiv.org/abs/2201.11903). This distinction in dataset size and composition is a key factor why our results are not directly comparable to those found in https://github.com/FranxYao/chain-of-thought-hub.
>
>
> **Question:** Hyperparameters chosen in our method
>
> **Answer:** Sorry for the typo, $\omega_2$ is $\beta$. The chosen of $\alpha$ and $\beta$ is from the grid search in [0, 1], the chosen of temperature for $T = 1.0$ is simply that is the largest allowed temperature value in the API.

---

### Author Response · Authors · 2023-11-16
**Responses to all reviewers (part 1)**

Thanks for the constructive comments, we address the following two frequently asked questions posed by reviewers. Additionally, we aim to highlight the following two questions that are crucial to the value of this paper's contributions.

**Question:** The authors can consider toning down the claims in the paper, for example: this paper primarily focuses on Question-Answering applications, but our same uncertainty estimates can also be applied to estimate how confident the LLM is in its response to a more general prompt.

**Answer:** In our study, we concentrate exclusively on Question-Answer scenarios because their performance is easily measurable. Moreover, our approach has other applications.

**For real company:** 1. We applied our method to assist attorneys in drafting documents. Our approach not only generates extensive text but also provides a confidence score for each draft. Here is one example (mask some confidential content):




`[Input]` [Some related document: prior to ... litigation and preserve them.]

`[Instruction]` Please provide a list of custodians, noncustodial data sources, and thirdparty data sources that may contain discoverable electronic information in connection with the instant litigation. In addition, please identify any inaccessible data sources that may contain discoverable information.

`[Response]` The answer to the question is not provided in the given text.

`[Confidence for response]` 0.042425017.



The response provided is incorrect; the relevant document does indeed have the necessary information. The attorney needs to re-examine this answer. Here is another example:


`[Input]`[Some related document: a producing ... linear review d.]

`[Instruction]` Please draft a clause that would allow a producing party to use tar to organize or prioritize documents for linear review without disclosure of that use so long as all documents identified by the electronic search methodology discussed above are reviewed as part of the linear review.

`[Response]` The producing party may utilize technology-assisted review (TAR) to organize or prioritize documents for linear review, ... without the obligation to disclose such use, provided that all documents identified by the electronic search methodology, as discussed above, are reviewed as part of the linear review process.

`[Confidence for response]` 0.939547988




Notably, response with low confidence scores are typically of low quality, suggesting that attorneys only need to review these to save considerable effort.


2. Additionally, we developed a chat window allowing interaction with our AI agent. In each conversation, the agent produces text accompanied by a confidence score. Here is one demo:

`>>` User: who was the only survivor of Titanic?

`>>` Agent: The only survivor of the Titanic who is still alive today is Millvina Dean. However, she passed away on May 31, 2009, at the age of 97.

`>>` **Confidence: 13.37%**

`>>` User: Are you sure Millvina Dean is the sole survivor of Titanic? Any other survivors?

`>>` Agent: No, Millvina Dean was not the sole survivor of the Titanic. There were 705 survivors in total out of the approximately 2,224 people on board. Millvina Dean was the youngest passenger on the ship and the last living survivor until her death in 2009. There were many other survivors, including notable individuals like Margaret Brown (also known as "The Unsinkable Molly Brown") and others. Charles Lightoller, who was the most senior officer to survive the sinking.

`>>` **Confidence: 93.44%**

From the first round conversation, the answer is wrong since over 700 people actually survived the catastrophe. The confidence level associated with this incorrect response was low. However, in the subsequent interaction, the correct information was supplied with a much higher confidence rating.

Given the challenge in quantifying the quality of long text generation, our main paper only focuses on question-answer generation. It's important to note that benchmark performance alone does not fully reflect a language model's capability. The true evaluation should consider the user experience during interactions between users and the agents. **Surveys of users of our chat application reveal that our method assigns useful confidence scores for long text generation in open-domain settings, and these users report the scores help them identify hallucinations.**

---

> ### Author Response · Authors · 2023-11-16
> **Responses to all reviewers (part 2)**
>
> **On the academic front**, the central issue in instruction tuning lies in being **data-centric**, focusing on how to effectively utilize high-quality data to fine-tune the large language model. We utilize our method to assign confidence scores for the Alpaca dataset (https://huggingface.co/datasets/tatsu-lab/alpaca). This enables us to assign a confidence rating to each output for a given input, as demonstrated by the following example:
>
> `[Input]` None
>
> `[Instruction]` Suggest ways to reduce plastic waste in everyday life.
>
> `[Response]` One way to reduce plastic waste in everyday life is to make the switch to reusable items. This includes bringing a reusable water bottle instead of buying bottled water, investing in a reusable coffee cup, bringing your own bags when shopping, and using reusable containers
> for storing and transporting food. You can also reduce the amount of plastic waste you create by avoiding or limiting the use of take-out or single-use containers, utensils, and straws. Additionally, try to find sustainable alternatives to plastic items when shopping and avoid purchasing items wrapped in plastic whenever possible. Finally, make sure you recycle any plastic items you do have.
>
> `[Confidence for Response]`: 0.9393
>
>  Here is another example:
>
> `[Input]` Banana
>
> `[Instruction]` Classify the item as either animal or vegetable.
>
> `[Response]` Animal: No, it’s a vegetable.
>
> `[confidence for Response]` 0.1099
>
> Our method distinguishes between high-quality and low-quality examples in the original datasets. By filtering out the low confidence examples and using only the high confidence ones, we fine-tune the Llama model. Here are the results of the fine-tuned LLama model:
>
> |        |   Alpaca with filtering data  |   Alpaca with original data   |
> |-----------|-------|-------|
> | Alpaca-eval win rate |   28.34%  | 26.46%  |
>
>
> We selectively fine-tune using only high-confidence data, removing any low-quality information. As demonstrated, this approach yields performance on par with or even superior to that achieved with the complete original alpaca dataset, as evaluated by the alpaca-eval metric.
>
> To conclude, we highlight several applications of our method for both industry and academic sides beyond its use in question-answer generation tasks.
>
>
> **Question:** Relationship to related work and novelty is missing.
>
> **Answer:** We want to point out here it is very challenge to benchmark some LLM baselines. This difficulty often arises due to the extensive use of meticulously crafted prompt engineering in some papers. Certainly, we will draw comparisons with Chain of Thought (CoT) prompting as we consider to be highly generalized methods that can be applied to a wide range of scenarios.
>
> Our goal is not necessarily to outperform some baselines, **as prompt engineering can significantly impact results**. Instead, our goal is to **develop a general-purpose pipeline across as many tasks as possible**, as the first question answering. We strive to minimize task-specific, ad-hoc processes in favor of creating simple, yet effective methods.
>
>
>
> But from the perspective of paper writing, we will discuss more about the related work in the revised version like reviewers suggested. For example, [1] prompts LLMs to generate both an answer and a level of confidence, but it is required to train additional models via supervised learning, unlike our method which does not employ any additional training. [2] introduces semantic entropy, incorporating linguistic invariances created by shared meanings. However their approach requires access to token-level probabilities from the LLM, which is often not accessible with today’s black-box APIs.
>
> [1] Stephanie Lin, Jacob Hilton, and Owain Evans. Teaching models to express their uncertainty in words. arXiv preprint arXiv:2205.14334, 2022a.
> [2] Lorenz Kuhn, Yarin Gal, and Sebastian Farquhar. Semantic uncertainty: Linguistic invariances for uncertainty estimation in natural language generation. arXiv preprint arXiv:2302.09664, 2023.

---

### Author Response · Authors · 2023-11-23

Dear Reviewer,

We haven't heard from you since sending you our rebuttal. Since we are approaching the last day of the reviewer-author discussion, it would be really nice of you to confirm if your initial concerns (most of them are clarification questions) were successfully addressed by our rebuttal. We are looking forward to your feedback and we kindly expect that you can raise the score if all your main concerns have been resolved.

Thanks!

Best regards,

Authors